# Mapping Rice Paddy Based on Machine Learning with Sentinel-2 Multi-Temporal Data: Model Comparison and Transferability

**Weichun Zhang [1], Hongbin Liu [1,*], Wei Wu [2], Linqing Zhan [3] and Jing Wei [4,5]**

[1] College of Resources and Environment, Southwest University, Chongqing 400716, China; love960223@email.swu.edu.cn

[2] College of Computer and Information Science, Southwest University, Chongqing 400716, China; ww@swu.edu.cn

[3] Chongqing Agricultural Technology Extension Station, Chongqing Municipal Committee of Agriculture and Rural Affairs, Chongqing 400121, China; zx123456zx@email.swu.edu.cn

[4] State Key Laboratory of Remote Sensing Science, College of Global Change and Earth System Science, Beijing Normal University, Beijing 100000, China; weijing@umd.edu

[5] Department of Atmospheric and Oceanic Science, Earth System Science Interdisciplinary Center, University of Maryland, College Park, MD 20740, USA

\* Correspondence: lhbin@swu.edu.cn

**Abstract:** Rice is an important agricultural crop in the Southwest Hilly Area, China, but there has been a lack of efficient and accurate monitoring methods in the region. Recently, convolutional neural networks (CNNs) have obtained considerable achievements in the remote sensing community. However, it has not been widely used in mapping a rice paddy, and most studies lack the comparison of classification effectiveness and efficiency between CNNs and other classic machine learning models and their transferability. This study aims to develop various machine learning classification models with remote sensing data for comparing the local accuracy of classifiers and evaluating the transferability of pretrained classifiers. Therefore, two types of experiments were designed: local classification experiments and model transferability experiments. These experiments were conducted using cloud-free Sentinel-2 multi-temporal data in Banan District and Zhongxian County, typical hilly areas of Southwestern China. A pure pixel extraction algorithm was designed based on land-use vector data and a Google Earth Online image. Four convolutional neural network (CNN) algorithms (one-dimensional (Conv-1D), two-dimensional (Conv-2D) and three-dimensional (Conv-3D_1 and Conv-3D_2) convolutional neural networks) were developed and compared with four widely used classifiers (random forest (RF), extreme gradient boosting (XGBoost), support vector machine (SVM) and multilayer perceptron (MLP)). Recall, precision, overall accuracy (OA) and F1 score were applied to evaluate classification accuracy. The results showed that Conv-2D performed best in local classification experiments with OA of 93.14% and F1 score of 0.8552 in Banan District, OA of 92.53% and F1 score of 0.8399 in Zhongxian County. CNN-based models except Conv-1D provided more desirable performance than non-CNN classifiers. Besides, among the non-CNN classifiers, XGBoost received the best result with OA of 89.73% and F1 score of 0.7742 in Banan District, SVM received the best result with OA of 88.57% and F1 score of 0.7538 in Zhongxian County. In model transferability experiments, almost all CNN classifiers had low transferability. RF and XGBoost models have achieved acceptable F1 scores for transfer (RF = 0.6673 and 0.6469, XGBoost = 0.7171 and 0.6709, respectively).

**Keywords:** rice; convolutional neural network; F1 score; sentinel-2; transfer

---

## 1. Introduction

Food security has become a global and international topic [1,2]. Rice, as one of the most important staple foods, is widely grown in Southern China. However, rice agriculture in China faces great challenges in the coming decade. The growing population has triggered an increase in the demand for rice [3]. Additionally, the acceleration of urbanization may create continual pressure in the rice cultivation area [4]. Other adverse factors such as droughts, soil degradation and economic restructuring should not be overlooked when studying rice production patterns [5]. Rice cultivation plays an important role in water resources utilization as well, since about a quarter to one-third of the world's freshwater resources are used for rice irrigation [6]. Rice paddies have been identified as an important source of methane ($CH_4$), which has a significant impact on the global greenhouse effect [7]. Therefore, it is essential to map the spatial distribution and planting area of paddy rice at a large scale for guiding rice production, water utilization, climate change and government policy decisions.

Traditionally, the large-scale rice mapping, which relies on massive ground surveys, is time-consuming, laborious and uneconomical. Satellite remote sensing data can provide timely, objective and accurate agricultural information at a regional to global scale [8,9]. Optical data are the most common and easily available satellite imagery data, and have found wide applications in spatial distribution extraction [10–12], cropping intensity [13] and the phenology [14] of a rice paddy. The identification of a rice paddy usually requires spectral bands, spectral indices (e.g., the land surface water index (LSWI) [15], normalized difference vegetation index (NDVI) [16], modified normalized difference water index (MNDWI) [17] and enhanced vegetative index (EVI) [18]) or their combinations, which are sensitive to the differences of spectral features between rice paddy and other classes [16,19]. However, the direct distinction of rice and other objects based on a certain phenological phase may be undesirable due to the spectral similarity of the different objects [20]. Extensive research has confirmed the effectiveness of multi-temporal analysis in rice paddy mapping during the specific growth phases [21–23]. For example, a pixel- and phenology-(the flooding/open-canopy) based rice growing area extraction algorithm was successfully tested in the Sanjiang Plain, China [21]. Cui et al. used NDVI, EVI and LSWI indexes during flooding/transplanting and ripening phases to identify rice fields, the user and producer accuracies of rice paddy were 90% and 94%, respectively [22].

Over the last decades, MODIS [16,18], Landsat TM/ETM+ [19,22] and GF-1 [24] have been usually used as a source of remote sensing image data in rice paddy mapping, but their spatial–temporal resolution is not sufficient to perform more detailed vegetation classification in areas with complex terrain. Launched by the European Space Agency (ESA) in 2015, the Sentinel-2 remote sensing satellite has a maximum spatial resolution of 10 m and a high frequency of 5 days at the equator [25]. Sentinel-2A optical data have been recently performed in crop classifications [26], water body mapping [27] and crop yield forecasting [28]. The excellent spatial and temporal resolution of the Sentinel-2 can ensure both the acquisition of critical phenological images of rice and the production of more accurate rice maps.

In recent years, many machine learning algorithms, such as decision trees (DTs) [29], support vector machines (SVMs) [30], random forest (RF) [30] and artificial neural networks (ANNs) [31], have been applied to map rice or other crop types. Currently, deep learning (DL) algorithms inspired by how the human brain recognizes and recalls information without outside expert input to guide the process have become a hot topic in the machine learning area [32]. One of the popular deep learning algorithms, convolutional neural networks (CNNs), had exhibited excellent performance levels in computer vision related contests such as ImageNet [33]. CNNs have been applied by the remote sensing (RS) community and shown advantages in pixel-based classification [19,34–38], object detection [39], semantic segmentation [40] and other tasks. Regarding pixel-based classifier, CNNs generally perform better than traditional machine learning algorithms. For example, Zhang et al. [19] showed that the Kappa coefficients of CNN are 7% and 9% higher than SVM and RF classifiers, respectively. CNNs received highest classification accuracy of 91% in the extraction of urban built-up areas [41].

Most traditional machine learning algorithms extract low-level features from the original images (e.g., spectral, terrain and texture). However, using convolutional kernels and the max-pooling method, CNNs can automatically extract high-level features from the original data [42,43], which are more robust and secure for the classification task and thus improve classification accuracy. There are many studies on crop classification using CNNs [19,42,44] according to the following steps: (1) building a dataset by a $w \times w$ ($w$ is the size of window) neighboring region from the original image, (2) extracting high-level features from the database by convolution on the spectral, the spatial or the spectral-spatial directions, (3) hierarchical model training and (4) model testing and classification.

Generally, there have been quite a few previous rice paddy or other land cover types classification efforts where various machine learning algorithms were leveraged [19,34–38,41,44]. For example, Zhang et al. [19] used the convolutional neural network to map the spatial distribution of rice in the Dongting Lake Area, China. Zhong et al. [37] utilized remotely sensed time series to classify crops in Yolo County, California based on machine learning models (including RF, SVM, CNN, etc.). Chen et al. [36] extracted urban water bodies from high-resolution remote sensing images based on deep learning. However, most studies lack the comparison of classification effectiveness and efficiency between CNNs and other traditional machine learning algorithms and their transferability.

The main goals of the current study were to (1) explore the feasibility of models for mapping rice paddy with massive labeled data and (2) evaluate the transferability of pretrained classifiers from one study area to another. Specifically, the rice area extraction models were constructed and compared by convolution on three different directions (spectral, spatial and spectral-spatial (both spectral and spatial)) by using Sentinel-2 multi-temporal image dataset in two study areas. Meanwhile, four popular non-CNN models were used for comparison. Finally, the best classification model was selected for mapping a rice paddy.

## 2. Study Area and Materials

### 2.1. Study Area

Our two study areas were the Banan district (106°26′–106°59′E, 29°7′–29°45′N) and the county of Zhongxian (107°3′–108°14′E, 30°3′–30°35′N) located in the hilly areas of Chongqing, Southwestern China (Figure 1). Elevations of Banan District and Zhongxian County vary between 154 and 1132 m, and between 117 and 1680 m, respectively. Both of them have a subtropical humid climate. The average annual temperature of Banan District and Zhong County is about 18.7 °C. Annual total precipitation is between 1000 and 1200 mm, which is mainly concentrated in May–July. The frost-free period is more than 300 days. Rice is the main food crop with similar phenology in both regions. Each year, rice is transplanted in April, heading and flowering in July, and harvested in September.

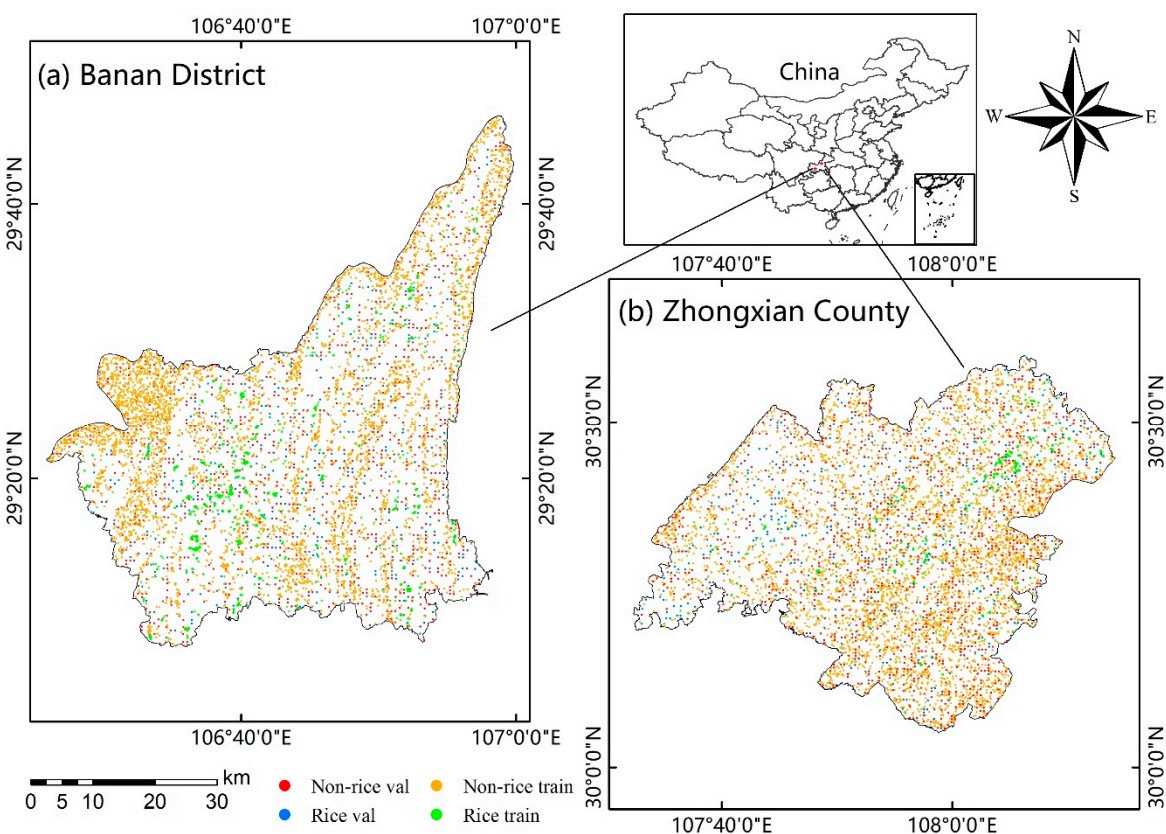

**Figure 1.** Maps of location, training samples and validation samples of Banan District (**a**) and Zhongxian County (**b**). "Non-rice val "and "Rice val" represent validation samples and "Non-rice train" and "Rice train" denote training samples (more detailed information please refer to supplementary (i.e., Figures S3 and S4)).

*2.2. Datasets*

2.2.1. Sentinel-2A Dataset

The available Sentinel-2A data were downloaded from Copernicus Open Access Hub (https://eros.usgs.gov/sentinel-2). The cloud-free (<2% cloud cover) images were acquired within rice growth season (April to October) in 2017. Finally, three images on 14 April 2017 (representing vegetative phase), 23 July 2017 (representing reproductive phase) and 31 October 2017 (representing post-harvest phase) were used for each study area, respectively. Sentinel-2 data were preprocessed using Sen2Cor plug-in and SNAP software, which mainly consists of atmospheric correction and resampling. The downloaded L1C-level top of the atmosphere (TOA) reflectance data were converted into L2A-level bottom of the atmosphere (BOA) reflectance data by the ESA's Sen2Cor plug-in. Then Sentienl-2 data were resampled to a 10 m resolution by bilinear resampling. Subsequently the original 60 m resolution bands and red-edge bands were culled with SNAP. Finally, six bands (Band 2, Band 3, Band 4, Band 8, Band 11 and Band 12) were used in this work.

Additionally, four spectral indices including normalized difference vegetation index (NDVI), land surface water index (LSWI), modified soil adjusted vegetation index (MSAVI) and modified normalized difference water index (MNDWI) were calculated according to Equations (1)–(4). NDVI is a widely used vegetation index and it is susceptible to soil background noise in low vegetation-covered areas [16]. LSWI is sensitive to the radiation equivalent water thickness and water content of leaf [15]. MSAVI is a vegetation index proposed to reduce the effects of soil noise [45]. MNDWI strengthens

the characteristics of open water bodies, and effectively suppresses or even eliminates noise from buildings, vegetation and soil [17].

$$NDVI = \frac{B8 - B4}{B8 + B4} \tag{1}$$

$$LSWI = \frac{B8 - B11}{B8 + B11} \tag{2}$$

$$MSAVI = 0.5 \times \left( 2 \times (B8 + 1) - \sqrt{(2 \times B8 + 1)^2 - 8 \times (B8 - B4)} \right) \tag{3}$$

$$MNDWI = \frac{B3 - B11}{B3 + B11} \tag{4}$$

Therefore, a total of 30 features derived from remote sensing images of three periods were tested to map rice paddy over the study areas, and each period includes 6 spectral bands and 4 spectral indices.

### 2.2.2. Training Samples

In order to avoid the influence of mixed pixel samples on classification accuracy, pure pixel samples were selected by 1:10,000 land-use map in 2017 and Google Earth Online image (Google Earth Pro v7.3.2.5776). This land-use map is generated using the visual interpretation approach and is available from the Chinese Land Resources Bureaus (http://www.mnr.gov.cn/). Its first version was generated in 2008 and renewed every year by the local land resources bureaus [19,46,47]. The land-use map comes from the 2017 land use utilization database of Zhongxian County and Banan District, and the report accuracy is over 90%. Here, a grid is a pure pixel if the grid itself and the surrounding pixels within a certain window size belong to the same land-use type. First, the land-use polygons were converted to raster format with a resolution of 10 m by ArcGIS 10.3. Second, consider a $w \times w$ square grid window such that the current pixel in consideration is at the center of it. The current pixel is considered "pure" if all $w \times w$ pixels in the grid have the same land cover category as the current pixel (at the center). This step was implemented in GDAL (Geospatial Data Abstraction Library, version 2.2.2) and python scripting language. Refer to the supplementary (i.e., Sections 1.1 and 1.2) for details of this process.

A value of $w$ that is too large might only produce samples distributed in large plots, while the w value that is too small will reduce the accuracy of pure pixel sample extraction. In Banan District and Zhongxian County, the length and width of the paddy field is usually between 50 and 100 m (representing 5–10 pixels at a 10-m spatial resolution). Therefore, $w$ was fixed at 7. After manual correction based on Google Earth Online image, 6359 pure pixels including 1559 rice and 4800 non-rice grids were obtained in Banan District. In Zhongxian County, 6400 pure pixels were obtained in the same way including 1600 rice and 4800 non-rice grids. These data were used as training samples for model construction (Table 1).

**Table 1.** The division of pure pixel samples in Banan District and Zhongxian County.

| | Banan District | | Zhongxian County | |
|---|---|---|---|---|
| | **Training** | **Validation** | **Training** | **Validation** |
| Rice | 1559 | 386 | 1600 | 386 |
| Non-rice | 4800 | 1189 | 4800 | 1180 |
| Total | 6359 | 1575 | 6400 | 1566 |

### 2.2.3. Validation Samples

Validation data for this study was assembled by manual inspection of Google Earth satellite images. First, we randomly distributed a set of points with a minimum spacing of 490 m between these points. Then, these points were classified as Rice/Non-Rice by manual inspection via the Google earth interface (imagery provided by TerraMetrics, CNES, etc.). Points that had very ambiguous land

covers were omitted. Finally, 1575 pixels in Banan District and 1566 pixels in Zhongxian County were generated for the validation of resultant paddy rice map (Figure 1). Table 1 shows the sample division of the two study areas.

## 3. Method

### 3.1. CNN Classifiers

Deep neural networks are composed of a number of hidden layers and are divided into full-connected neural network/multilayer perceptron (FC/MLP) convolutional neural networks (CNNs) and recurrent neural networks (RNNs). CNNs have recently become a popular deep neural networks and have achieved good success in computer vision and image understanding such as ImageNet contest. In convolutional neural networks, every image input is treated as a matrix of pixel values, which represents the gray values at a given pixel in the image. Unlike traditional neural networks, which treat an image as a one-dimensional network, the CNNs consider the spatial relevance between each pixel and its neighboring pixels. CNNs consist of the following layers: A. Input layer, B. Convolutional layer, C. Activation layer, D. Pooling layer, E. Full-connected layer and F. Output layer. The meaning and calculation formula of each layer can refer to [48], the following is only a brief introduction.

A. Input layer: Used for data input.
B. Convolutional layer: In this layer, a kernel (or weight) matrix is used to extract high level features from the images. The kernel with its weights rotates over the image matrix in a sliding window in order to obtain the convolved output. The kernel matrix behaves like a filter in an image extracting particular information from the original image matrix. During the convolution process, the weights are learnt such that the loss function is minimized.
C. Activation layer: The role of the activation layer is to improve the expression capacity of the entire network by increasing nonlinear mapping, and the rectified linear unit (ReLU) f (x) = max (x, 0) is commonly chosen to be an activation function.
D. Pooling layer: Pooling layers are used to extract the most informative features from the generated convolved output.
E. Full-connected layer: Generally located at the end of CNNs, connecting all features and outputting all values to the classifier.
F. Output layer: To generate the final output.

According to the dimension of the convolutional filter, CNNs may be categorized into one-dimensional convolutional neural network (Conv1D), two-dimensional convolutional neural network (Conv2D) and three-dimensional convolutional neural network (Conv3D). Previous studies mostly focused on Conv1D and Conv2D [19,43,44,49]. In this work, Conv1D and Conv2D as well as Conv3D classifiers are constructed to map rice paddy and they are briefly described below:

A. Conv-1D classifier

Conv1D, which is a most simplistic form of CNNs, is often used for sequence datasets. This model can be used for extracting local one-dimensional subsequences from the input sequences and identify local patterns within the window of convolution. Spectral information of each pixel is considered in this case. Figure 2a shows the structure of Conv1D, which consists of two convolutional layers, one flatten layer, two full-connected layers, as well as one batch-normalization layer and one dropout layer to prevent overfitting. The popular activation function called rectified linear unit (ReLU) is used to increase the nonlinear expression of neural networks and solve the vanishing gradient problem.

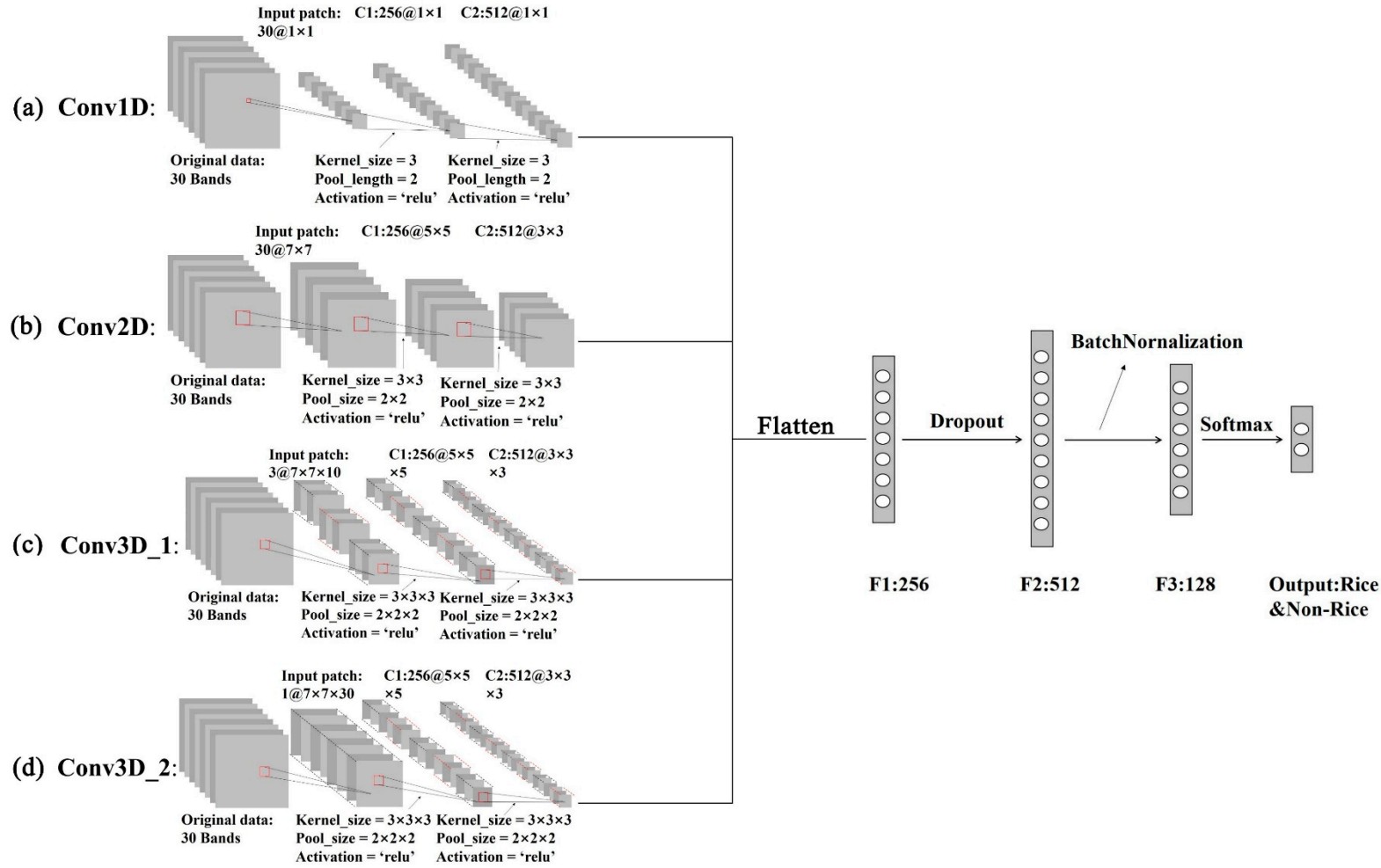

**Figure 2.** Network structure of Conv1D (**a**), Conv2D (**b**), Conv3D_1 (**c**) and Conv3D_2 (**d**). The input layer, fully connected layer and output layer of the four classifiers remain unchanged. Convolution kernels and pooling kernels have different dimensions, which vary according to the convolution dimensions. The parameters of convolutional neural networks (CNNs) are referenced at "https://keras.io/".

B.     Conv-2D classifier

Two-dimensional convolutional filters are widely used for image analyses. The main idea of Conv-2D is that the convolutional filter moves in 2-directions (X, Y) to calculate higher-level features from the set of lower-level feature maps. In contrast to the Conv1D, the spatial relationships between pixels rather than spectral information are calculated. In the case of Sentinel-2 (10-m spatial resolution), we have found that the window size of $7 \times 7$ is able to capture spatially local correlation of a center pixel to the surrounding pixels and limit heterogeneous pixels. Figure 2b illustrates the structure of Conv-2D model. The model includes one flatten layer, two convolutional layers, two maxpooling2D layers and two dense layers. To prevent overfitting, one dropout and one batch-normalization layers are added. Conv-2D classifier also uses ReLU as the activation function.

C.     Conv-3D classifier

In this section, a CNN with 3D convolution is designed for both spatial and spectral features. A three-dimensional kernel is applied to the cube formed by stacking multiple contiguous feature maps together and moved in three-directions (X, Y and Z) to calculate the high-level feature representations. Figure 2c,d show two Conv-3D architectures (Conv-3D_1 and Conv-3D_2). The inputs (X, Y, Z, Channel) of the two models are different, where X and Y are the size of window, which is consistent with the size of the convolution kernel in Conv-2D, Z is the number of bands for a cube and Channel is the number of cubes. Conv-3D_1 uses three cubes according to three periods, each cube having 10 bands. Conv-3D_2 employs one cube that is composed of 30 features. The neural network structures of the two models are the same except for the input layer, which consists of one flatten layer, two convolutional layers, two full-connected layers and several overfitting layers.

*3.2. Other Classifiers*

Four non-CNN classifiers including random forest (RF), extreme gradient boosting (XGBoost), support vector machine (SVM) and multilayer perceptron (MLP) are compared with CNNs for mapping a rice paddy. These classification algorithms are known for high performance and have been widely applied in remote sensing community [30,31,50]. RF is an ensemble learning algorithm for the classification and regression based on the bagging technique, which is suitable for processing high-dimensional data and preventing overfitting [50,51]. XGBoost is an extensible end-to-end boosting system, which is widely used in many machine learning competitions and designed as the main component of solutions [52]. SVM is a discriminant classifier for separating hyperplane, in which the input vectors are nonlinearly mapped into a very high dimensional feature space by using kernel functions [53]. MLP is a plain deep neural networks, in which neurons of a layer are fully connected with all neurons of neighboring layers [54].

The models are developed using the Scikit-learn and XGBoost packages established in Python. A grid search method embedded in Python (GridSearchCV) was applied to optimize the models' parameters. The optimal parameter combinations for RF, SVM, XGBoost and MLP are given in Table 2. The models were executed on a computer with Intel Core i5-9400F processor, NVIDIA GeForce RTX 2060 (6 GB), and RAM 8 Gb. CNN classifiers were run using Python v3.6.6, Tensorflow-gpu v1.13.1 and Keras v2.2.4.

**Table 2.** Parameters of non-CNN classifiers.

| Algorithm | Abbreviation | Hyperparameter | Candidate Values | Number of Combinations | Optimal Parameter Value |
|---|---|---|---|---|---|
| Random Forest | RF | n_estimators<br>max_features | {(100,210),10}<br>2,3,4,5,6,7,8 | 70 | 180<br>7 |
| Support Vector Machine | SVM | C<br>gamma | $10^{-2}, 10^{-1}, 1.0, 10, 10^2, 10^3$<br>0.01,0.1, 1, 2, 8,10 | 75 | 10<br>10 |
| Extreme Gradient Boosting | XGBoost | reg_alpha<br>colsample_bytree<br>subsample | 0.001, 0.01, 0.1<br>0.5, 0.6, 0.7, 0.8, 0.9<br>0.5, 0.6, 0.7, 0.8, 0.9 | 36 | 0.1<br>0.5<br>0.7 |
| Multilayer Perception | MLP | learning_rate_init<br>batch_size | $10^{-5}, 10^{-4}, 10^{-3}, 10^{-2}, 10^{-1}$<br>{(20, 120), 20} | 35 | 0.001<br>80 |

### 3.3. Model Performance

The classifier performance was evaluated by the overall accuracy (OA), recall, precision and F1 score calculated from the confusion matrix (Table 3). Overall accuracy is a widely used indicator in classification, which represents the ratio between the model and the total number predicted on all test sets. Precision is the fraction of retrieved instances that are relevant, while recall is the fraction of relevant instances that are retrieved. For original sample and predicting results, recall and precision are the ratio of the correct predictions and the total number of actual items or predicted items in the set, respectively. Generally, precision and recall are a pair of contradictory measures, when one is higher, the other is often lower. The F1 score is the harmonic average of the precision and recall with 1 indicating the best and 0 denoting the worst.

**Table 3.** Confusion matrix.

| | | Prediction | | |
|---|---|---|---|---|
| | | Rice | Non-Rice | Sum |
| **Ground Truth** | **Rice** | True Positive (*TP*) | False Negative (*FN*) | Actual Positive (*TP* + *FN*) |
| | **Non-Rice** | False Positive (*FP*) | True Negative (*TN*) | Actual Negative (*FN* + *TP*) |
| | **Sum** | Predicted Positive (*TP* + *FP*) | Predicted Negative (*FN* + *TN*) | *TN* + *TP* + *FN* + *FP* |

The OA, recall, precision and F1 score were calculated using Equations (5)–(8), respectively:

$$OA = \frac{TP + TN}{TN + TP + FN + FP} \tag{5}$$

$$Recall = \frac{TP}{TP + FN} \tag{6}$$

$$Precision = \frac{TP}{TP + FP} \tag{7}$$

$$F1\ score = \frac{2 \times Recall \times Precision}{Recall + Precision} \tag{8}$$

## 4. Results and Analysis

### 4.1. Model Performance at Each Site

Two multi-temporal data sets (30 bands total) were used to train and test CNNs and non-CNN classifiers for the two study areas. Model performance was evaluated at each site (Table 4).

For CNN classifiers, F1 scores of Conv-1D, Conv-2D, Conv-3D_1 and Conv-3D_2 in Banan District were 0.7544, 0.8552, 0.8067 and 0.8196, respectively (Table 4a) and F1 scores of these classifiers in Zhongxian County were 0.7451, 0.8399, 0.8260 and 0.8321, respectively (Table 4b). In Banan District, Conv-2D gave the highest values of recall (0.8264), OA (0.9314) and F1 score (0.8552) while Conv-3D_2

yielded the highest value of precision (0.8906). In Zhongxian County, Conv-2D achieved the highest values of precision (0.8899), OA (0.9253) and F1 scores (0.8399) while Conv-1D gave the highest values of recall (0.9585). However, Conv-2D had the lowest values of recall (0.7953). Besides, Conv-2D and Conv-3D_2 presented similar values of F1 score, but Conv-2D had a lower training time (230 s) and testing time (2320 s) than Conv-3D_2 (training time = 333 s and testing time = 3626 s). Moreover, both in Banan District and Zhongxian County, Conv-1D had the lowest values of precision (0.7827 and 0.5929), OA (0.8838 and 0.8340) and F1 score (0.7544 and 0.7451), indicating that the convolution in the spectral direction only might be unsuitable for rice planting region extraction over the study site.

**Table 4.** Classifiers local accuracy evaluation in Banan District (**a**) and Zhongxian County (**b**); Boxes with black and grey colors represent the maximum and minimum values of the evaluation indicators of CNNs or non-CNNs, respectively).

| | | | | (a) Banan District Local Classification Results | | | |
|---|---|---|---|---|---|---|---|
| | Classifier Type | Recall | Precision | Overall Accuracy | F1 Score | Training Time (s) | Testing Time (s) |
| CNNs | Conv-1D | 0.7280 | 0.7827 | 0.8838 | 0.7544 | 57 | 680 |
| | Conv-2D | 0.8264 | 0.8861 | 0.9314 | 0.8552 | 127 | 1030 |
| | Conv-3D_1 | 0.7668 | 0.8709 | 0.9117 | 0.8067 | 1210 | 14,948 |
| | Conv-3D_2 | 0.7591 | 0.8906 | 0.9177 | 0.8196 | 443 | 4378 |
| | Mean | 0.7701 | 0.8576 | 0.9112 | 0.8090 | | |
| Non-CNNs | RF | 0.6865 | 0.8632 | 0.8960 | 0.7648 | 0.7662 | 759 |
| | XGBoost | 0.7150 | 0.8440 | 0.8973 | 0.7742 | 0.4933 | 113 |
| | SVM | 0.6658 | 0.8624 | 0.8916 | 0.7515 | 1.3056 | 1963 |
| | MLP | 0.4482 | 0.9402 | 0.8571 | 0.6070 | 203.48 | 380 |
| | Mean | 0.6268 | 0.8775 | 0.8855 | 0.7244 | | |
| Overall mean | | 0.6995 | 0.8675 | 0.8984 | 0.7667 | | |
| | | | | (b) Zhongxian County local classification results | | | |
| | Classifier Type | Recall | Precision | Overall Accuracy | F1 score | Training Time (s) | Testing Time (s) |
| CNNs | Conv-1D | 0.9585 | 0.5929 | 0.8340 | 0.7451 | 127 | 755 |
| | Conv-2D | 0.7953 | 0.8899 | 0.9253 | 0.8399 | 230 | 2320 |
| | Conv-3D_1 | 0.8238 | 0.8281 | 0.9144 | 0.8260 | 551 | 7878 |
| | Conv-3D_2 | 0.8472 | 0.8175 | 0.9157 | 0.8321 | 333 | 3626 |
| | Mean | 0.8562 | 0.7821 | 0.8973 | 0.8108 | | |
| Non-CNNs | RF | 0.6917 | 0.8042 | 0.8825 | 0.7437 | 0.7285 | 759 |
| | XGBoost | 0.7228 | 0.7815 | 0.8819 | 0.7510 | 0.5123 | 113 |
| | SVM | 0.7098 | 0.8035 | 0.8857 | 0.7538 | 1.3056 | 1963 |
| | MLP | 0.7539 | 0.7293 | 0.8704 | 0.7414 | 156.4 | 380 |
| | Mean | 0.7196 | 0.7796 | 0.8801 | 0.7475 | | |
| Overall mean | | 0.7879 | 0.7809 | 0.8887 | 0.7791 | | |

For non-CNN classifiers, in Banan District, the XGBoost classification model showed the best performance indicated by both accuracy and operation time. This model has the highest recall (0.7150), overall accuracy (0.8973) and F1 score (0.7742) and the lowest training time (0.4933 s) and testing time (113 s). RF and SVM classifiers also gave higher classification accuracy than MLP classifier with F1 scores of 0.7648 and 0.7515, respectively. Besides, the RF classifier was more efficient than the SVM classifier. The testing time of the RF classifier was 759 s, which was much lower than that of SVM (1963 s). MLP model presented the lowest values of recall (0.4482), overall accuracy (0.8571) and F1 score (0.6070) and the highest value of training time (203.48 s). In Zhongxian County, the SVM classifier gained the highest overall accuracy (0.8857) and F1 score (0.7538). MLP presented the lowest accuracy value of precision (0.7293), OA (0.8704) and F1 score (0.7414) and the highest value of precision (0.7539). Besides, RF presented the lowest value of recall (0.6917) and the highest value of precision (0.8042).

On average, CNN classifiers produced higher values of recall, precision, OA and F1 scores than non-CNNs. Among the evaluated methods, the Conv-2D model had the highest accuracy indicated by OA and F1 score. Besides, the operation time (training time and testing time) of the Conv-2D classifier was also acceptable. The two maps of rice paddy of Banan District and Zhongxian County produced by the Conv-2D classifier are shown in Figure 3a,b. In order to compare the detailed mapping information created by the evaluated models, two subregions were selected randomly and illustrated in Figure 4.

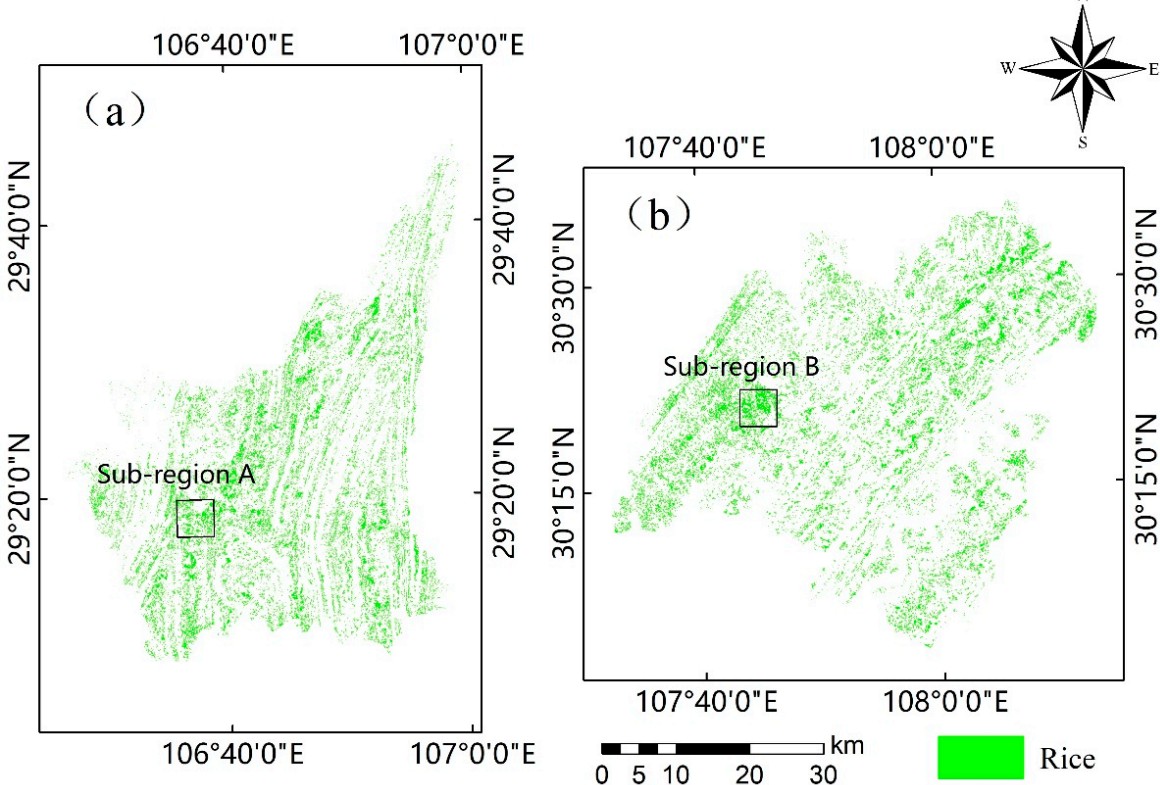

**Figure 3.** Paddy-rice map of the study area generated by Conv-2D in Banan District (**a**) and Zhongxian County (**b**) the two subregions with 5 km × 5 km were used to compare the detailed information created by the models).

Figure 4a,b shows two subregions (5 km × 5 km) of rice paddy maps generated by the eight classifiers in Banan District (Subregion A) and Zhongxian County (Subregion B), respectively. To give a better visual comparison, intersect and erase calculations were performed using ArcGIS on the reference land use map and our classification result. Green color represents the overlap areas between our classification and the reference maps, and red and yellow colors represent missing classification pixels (i.e., actual rice pixels were not identified) and wrong classification pixels (i.e., actual non-rice pixels were identified as paddy pixels), respectively.

Non-CNN classifiers generated massive isolated rice pixels. These pixel-based classification results tended to possess "salt and pepper" noise. Besides, the classification performance of MLP was unstable, which may seriously underestimate the distribution of rice areas in Banan District, but performs well in Zhongxian County. The classification maps of Conv-2D, Conv-3D_1 and Conv-3D_2 were similar. Moreover, compared with the other models, Conv-1D greatly overestimated the distribution of rice planting areas in Subregion B. Furthermore, the classification maps of Conv-2D, Conv-3D_1 and Conv-3D_2 were relatively smoother than that of non-CNNs and a Conv-1D classifier. The main reason may be the use of patch-based method, which selects the image block centered on the target pixel as the input of the classifier, and the spatial associations and semantic links between pixels in the image block are helpful to produce homogeneous and smooth images. Unfortunately, eight classifiers did not perform well in the extraction of narrow rice paddies. Although Conv-2D, Conv-3D_1 and Conv-3D_2 achieved sound classification results but could not accurately identify overly narrow rice paddies.

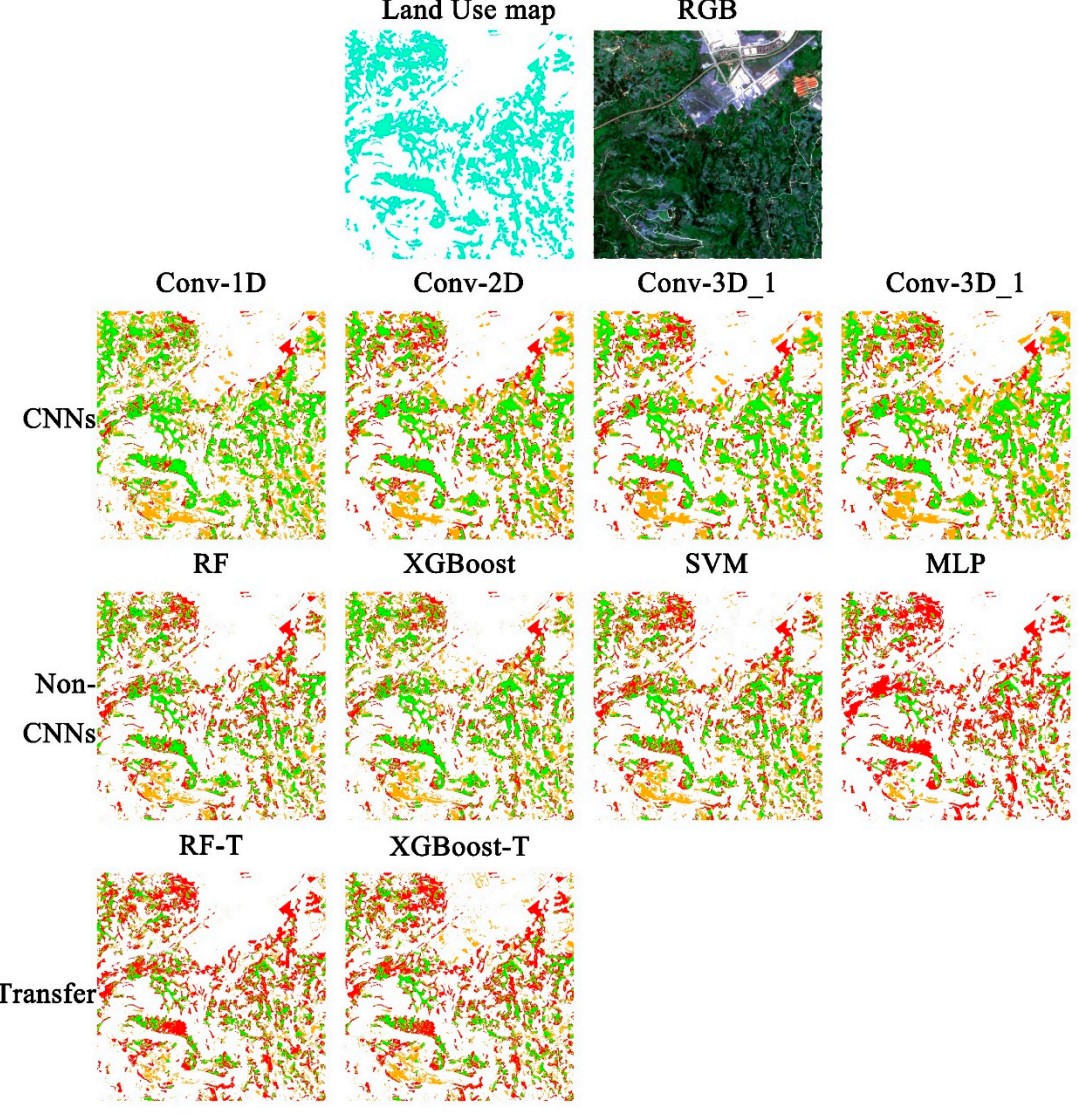

(**a**) Subregion A

**Figure 4.** *Cont.*

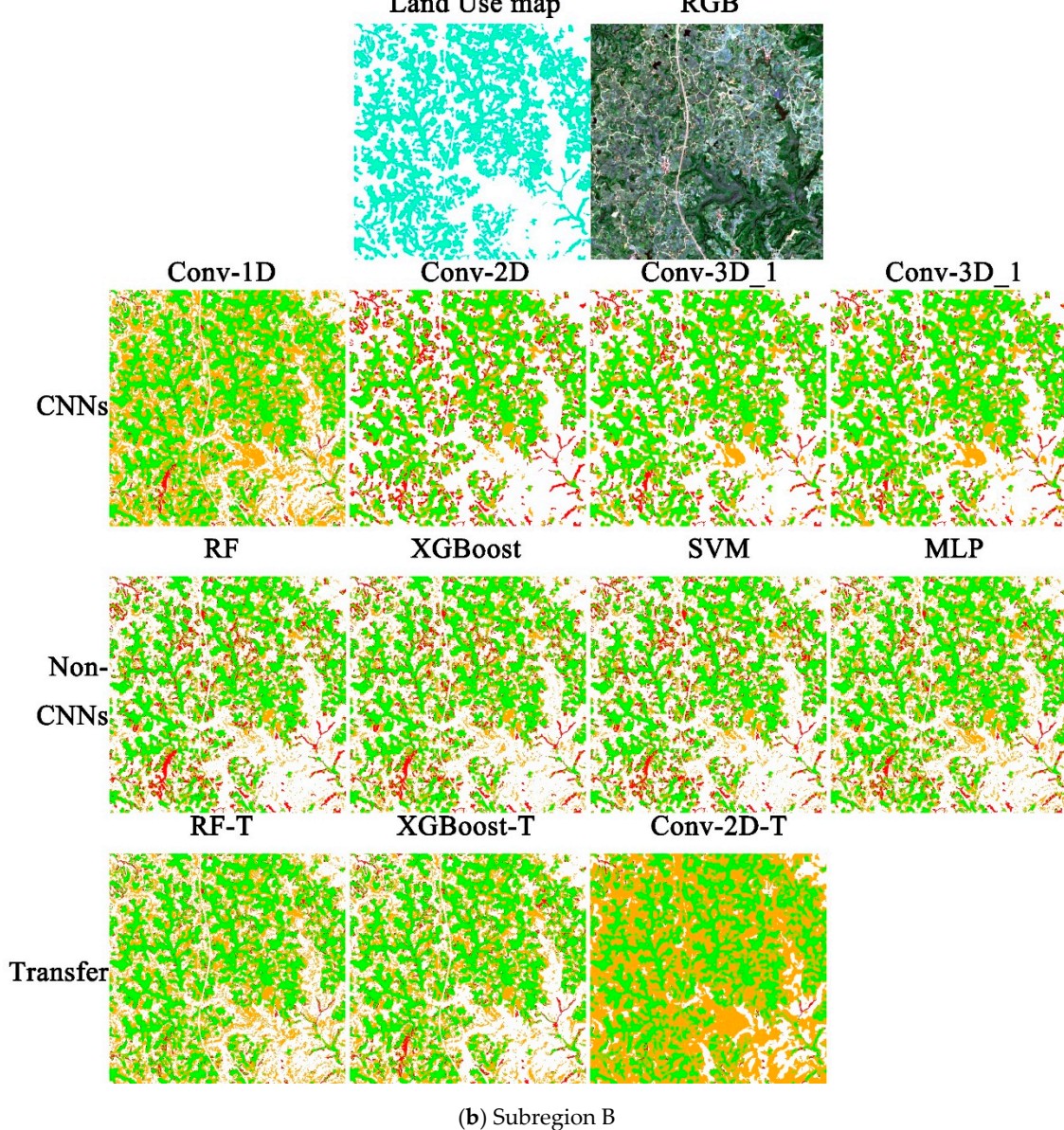

(**b**) Subregion B

**Figure 4.** Subregion classification results ((**a**) Subregion A in Banan District and (**b**) Subregion B in Zhongxian County) based on different methods, Land Use Map (cyan-blue pixels represent the referenced rice pixels) is land-use vector data in 2017 for comparison, RGB is Sentinel-2 true color composite image (green pixels represent the overlap between result maps and the reference map, and red pixels and yellow pixels represent under-classification pixels and over-classification pixels, respectively); transfer represents the transferable results of the relatively reasonable classifiers (refer to Section 4.2).

## 4.2. Model Transferability

To estimate the transferability of the machine learning classification models, two experiments (A and B) were conducted. Experiment A computed rice paddies in Zhongxian County using the trained classification models based on the dataset of Banan District. Experiment B extracts rice paddies in Banan District using the trained classification models based on the dataset of Zhongxian County. The transferability evaluation results are shown in Table 5.

**Table 5.** Classifiers transfer accuracy evaluation in Experiment A and Experiment B.

| (a) Experiment A: Classifiers that Trained in Banan District Transferred to Zhongxian County | | | | | |
|---|---|---|---|---|---|
| | **Classifier Type** | **Recall** | **Precision** | **Overall Accuracy** | **F1 Score** |
| **CNNs** | **Conv-1D** | 0.9845 | 0.4064 | 0.6418 | 0.5753 |
| | **Conv-2D** | 0.9637 | 0.4856 | 0.7395 | 0.6458 |
| | **Conv-3D_1** | 0.9974 | 0.2663 | 0.3218 | 0.4203 |
| | **Conv-3D_2** | 0.9870 | 0.3940 | 0.6226 | 0.5632 |
| **Non-CNNs** | **RF** | 0.8756 | 0.5391 | 0.7848 | 0.6673 |
| | **XGBoost** | 0.8109 | 0.6427 | 0.8423 | 0.7171 |
| | **SVM** | 0.0000 | - | 0.7535 | - |
| | **MLP** | 0.9974 | 0.2860 | 0.3857 | 0.4446 |
| (b) Experiment B: Classifiers that Trained in Zhongxian County Transferred to Banan District | | | | | |
| | **Classifier Type** | **Recall** | **Precision** | **Overall Accuracy** | **F1 Score** |
| **CNNs** | **Conv-1D** | 0.2228 | 0.6565 | 0.7800 | 0.3327 |
| | **Conv-2D** | 0.9741 | 0.2684 | 0.3399 | 0.4208 |
| | **Conv-3D_1** | 1.0000 | 0.2539 | 0.2768 | 0.4050 |
| | **Conv-3D_2** | 0.9689 | 0.2980 | 0.4305 | 0.4558 |
| **Non-CNNs** | **RF** | 0.5078 | 0.8909 | 0.8635 | 0.6469 |
| | **XGBoost** | 0.5466 | 0.8683 | 0.8680 | 0.6709 |
| | **SVM** | 0.0000 | - | 0.7538 | - |
| | **MLP** | 0.1088 | 1.0000 | 0.7806 | 0.1963 |

The classification models with reasonable F1 score of higher than 0.6 in Experiment A included Conv-2D (0.6458), RF (0.6673) and XGBoost (0.7171). Figure 4b shows that the number of yellow pixels in the result maps was larger than the reference map and maps produced by local models. Moreover, by comparing Tables 4 and 5, the transfer classification accuracy of Conv-2D, RF and XGBoost was lower than that of the local models. In Experiment B, XGBoost presented the highest F1 score of 0.6709 followed by RF (0.6469). However, the values of recall (0.5078, 0.5466) were much lower than the values of precision (0.8909, 0.8683), indicating that the transferred models ignored a large number of actual rice pixels (the number of red pixel increases compared to reference map and result maps produced by local models).

Almost all CNN classifiers had low transferability. According to the F1 score, the Conv-2D model gave the reasonable classification accuracy in Experiment A. However, a huge imbalance between the values of recall (0.9637) and precision (0.4856) indicates that the model recognizes a large number of non-rice pixels as rice paddies (Conv2D-T in subregion B having a large number of yellow pixels). For non-CNN models, SVM and MLP classification models showed low transferability in terms of the performance indicators. RF and XGBoost classification models could be transferred to other research areas, although the transferable accuracy was lower than the classification accuracy at the local site.

## 5. Discussion

### *5.1. Evaluation of Classifier's Local Classification Accuracy*

#### 5.1.1. Performance Comparison of CNNs

Deep learning algorithms, which learn the internal rules and presentation hierarchy of sample data, have outperformed most traditional machine learning in land-use or crop classifications [34,37,42,55]. In particular, as the most representative deep learning algorithm, CNNs have been widely used in remote sensing community and have achieved a good performance [19,34,35,42,44]. In this study, three CNN classifiers (Conv-2D, Conv-3D_1 and Conv-3D_2) that used the patch-based method gave higher accuracy than others. RF, XGBoost, SVM, MLP and Conv-1D are pixel-based, where land-use classes are assigned to each individual pixel allocated. For large-scale classification, the spatial context plays a major role and provides inherent information about the target pixels. It is difficult to adequately describe the feature information of the ground object with only the single-pixel independent spatial

information. In this case, the well-trained pixel-based classification models may not be robust to resist the salt-and-pepper noise and fragmentation, which was clearly shown in Figure 4a. The patch-based method produces image objects (patches) for model input. Spatial relationships and semantic links between pixels can be explored with the patch-based method, resulting in more homogeneous images. Prior studies have demonstrated the importance of patch-based methods [19,41]. Further, two-dimensional and three-dimensional convolutional neural networks have advantages in rice area extraction due to the use of patch-based methods. Zhang and Lin applied the CNN model to extract rice planting areas and the overall accuracy of the CNN is 6.43% and 7.68% higher than that of RF and SVM, respectively [19]. Similar results were obtained in South China with the overall accuracy of CNN and SVM being 93.60% and 91.05%, respectively [56]. CNN is also used in identifying and mapping of other geo-objects [57]. Two-dimensional convolutional neural networks have the highest classification accuracy of 91% for identifying urban built-up areas [41].

The use of convolution is another reason for improving the performance of Conv-2D, Conv-3D_1 and Conv-3D_2. Convolution extracts and creates complex features and multilevel convolution can ensure that the model has sufficient expressive and generalization ability [42,55,56,58]. Chen et al. evaluated the effectiveness of convolution by analyzing the intermediate activation feature maps for comparing the similarity in the same class and the divisibility between different classes [58]. They found that convolutional direction or scenario was closely related with model performance. In both experimental areas of the current work, Conv-1D obtained the lowest F1 score among the CNNs. This is in agreement with the result of Kestur, R., et al. [49]. However, Conv-1D has achieved good performance in land use classification based on single- or multi-temporal hyperspectral datasets [49,58] and vegetation index (NDVI, EVI, etc.) time-series data [37], for example, Chen et al. [58] and Guidici et al. [49] reported that the Conv1D architectures can be applied to hyperspectral data and achieve high classification accuracy. This is mainly because hyperspectral datasets used by Chen et al. [58] and Guidici et al. [49] have inherited sequential interdependence, the spectral features in a single pixel are closely linked, allowing for more beneficial features to be convolved in an integrated machine learning framework. However, the data used in that article have lower sufficient sequence dependencies. Convolution only in the spectral direction may introduce more noise. This could explain why Conv-3D_1 and Conv-3D_2 classifiers gave slightly less accurate results than Conv_2D, since Conv_3D_1 and Conv_3D_2 are convoluting simultaneously in the spectral and spatial directions.

### 5.1.2. Performance Comparison of Non-CNNs

XGBoost had the best classification performance among all the non-CNN classifiers in Banan District. XGBoost is considered by many scholars or contestants as one of the best algorithms for processing non-perceptual data [50,59]. Recently, XGBoost has obtained considerable success in remote-sensing data analysis [37,50,60]. Zhong used Landsat enhanced vegetation index (EVI) time series data for crop classification, and XGBoost achieved a better result than SVM and RF [37]. Joharestani used XGBoost, RF and MLP to predict the PM 2.5 distribution in Tehran's urban area and XGBoost obtained the best model performance with $R^2$ of 0.81 [50]. SVM produced the highest F1 score for mapping rice paddies among the four non-CNN classifiers in Zhongxian County. The efficient performance of SVM has been confirmed in some comparative studies [50,57]. Abdi used multi-temporal Sentinel-2 data for land cover and land use classification in complex boreal landscapes based on a variety of non-parametric algorithms. SVM achieved the highest overall accuracy of 0.758, followed by XGBoost of 0.751 [59]. However, SVM had the longest testing time in non-CNN classifiers, reflecting its inherent shortcomings of low operating efficiency when generalized to new samples outside the training dataset in large datasets. In the two local experiments of this paper, RF performed moderately and stably. Rodriguez-Galiano has verified the high performance and robustness of RF in land-use classification based on several criteria: sensitivity to data set size, mapping accuracy and noise [57].

## 5.2. Comparison of Model Transferability

Convolutional neural networks (except one-dimensional convolution) performed well at local areas, but worst at other sites. Advanced deep neural networks have achieved impressive results on many image classification tasks [19,34,35,48]. However, deep learning models are fragile and unstable, and their generalization ability has always been a challenge. In fact, subtle and imperceptible perturbations of the data samples are sufficient to fool deep learning classifiers and result in incorrect classification [61]. A classic study is to superimpose noise that is not perceivable by human eyes (which does not affect human recognition) on panda photos, but the neural network model identifies it into other categories [62]. In other words, the similarity between the training dataset and the testing dataset could severely affect the performance of deep learning-based methods, and if the new inputs differ from what the net saw at training time, the model might break in absurd ways. In the remote sensing community, studies have confirmed that models trained at one site may not be applicable to other areas due to radiometric variation of source and target datasets [63,64]. Moreover, CNN classifiers are based on patches, which extract high-dimensional features by inputting presegmented sample blocks and then perform model training. The sample blocks contain the spatial context information of the geo-objects, and the differences in the morphology of the geo-objects between the source domain and target domain might be detrimental to the model transfer. Rice paddies in Banan District are relatively narrow and scattered than that of Zhongxian County. Consequently, in this study, massive interference might reduce the transferability of models due to the radiometric variation and morphological difference between source and target datasets.

The tree-based ensemble learning algorithms, RF and XGBoost, performed well in the transfer experiments. Ensemble learning builds and combines multiple learners to complete learning tasks, and often can obtain significantly better generalization capabilities than individual learners (such as C4.5 decision tree algorithms, SVM, MLP, etc.) and reduce the risk of falling into a local optimum. Rodriguez-Galiano's research shows that the RF outperforms the decision tree in noise immunity [57]. The basic classifiers of RF and XGBoost are both decision trees, which subdivide the input data according to the condition judgment and have stronger anti-interference ability than SVM and MLP.

## 5.3. Deficiencies and Prospects

Conv-2D classifiers in this article gave satisfactory classification accuracy, however there might be numerous mixed pixels in the result maps. The future work is considering combining deep learning and spectral unmixing for subpixel-level land use/cover classification. Moreover, the potential of CNNs cannot be fully exploited because of the inconsistent size between the original image and the ever-decreasing feature image. Some deep or ensemble learning architectures that perform well in image semantic segmentation, such as U-Net [62], PSP-Net [65], and HRNet [66], and extremely randomized trees [67,68], could provide alternatives to extract geo-objects. Besides, how to improve the transferability of CNN models is worthy of in-depth research. Some studies have attempted to increase the transferability and generalization capabilities of models through transfer learning or data augmentation techniques. Data augmentation resamples the original input, including geometric and/or radiosity transformations to expand the sample space and improve the generalization ability of the models [63]. Transfer learning uses the knowledge learned from the source data domain to solve a new or related task in the target domain, which is expected to solve the problem of training reliable models without sufficient labeled data. A common used idea of transfer learning is to obtain the feature representations, which are close to the source data domain for model training in the target domain [69].

## 6. Conclusions

We developed deep-learning CNNs with various architectures and four popular non-CNN models to extract rice growing areas using multi-temporal sentinel-2 data in Banan District and Zhongxian County. The most satisfactory classification result was achieved by a two-dimensional convolutional

neural network (Conv-2D) in both study areas. According to F1 scores and overall accuracy, Conv-2D was significantly superior to popular Non-CNN classifiers such as random forest, XGBoost and support vector machine as well as one-dimensional convolutional neural network (Conv-1D). The proposed method gave overall accuracy and F1 scores of higher than 95% and 0.90, respectively. Moreover, among the non-CNN classifiers, SVM and XGBoost received the best result in Banan District and Zhongxian County, respectively. Besides, two experiments were designed to study the transferable performance of eight models. Almost all CNN classifiers had low transferability, while RF and XGBoost models achieved acceptable transferability in both experiments.

**Supplementary Materials:** The following are available online at http://www.mdpi.com/2072-4292/12/10/1620/s1, Figure S1: Array slicing algorithm; Figure S2: An example of pure pixel extraction procedure; Code based on python and GDAL; Figure S3: Training samples and validation samples of Banan District; Figure S4: Training samples and validation samples of Zhongxian County.

**Author Contributions:** Methodology, W.Z., H.L. and W.W.; Resources, L.Z.; Software, W.Z.; Supervision, H.L.; Writing—original draft, W.Z.; Writing—review and editing, H.L., W.W. and J.W. All authors have read and agreed to the published version of the manuscript.

**Funding:** This research received no external funding.

**Conflicts of Interest:** The authors declare no conflict of interest.

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
