# Peer review of "Mapping Rice Paddy Based on Machine Learning with Sentinel-2 Multi-Temporal Data: Model Comparison and Transferability"

_remotesensing, doi:10.3390/rs12101620_

Round 1

Reviewer 1 Report

The topic of the manuscript is interesting, and it is well structured. Models are well explained, and results are consistent with results already obtained by other authors. I suggest an English editing by a native speaker in order to improve the manuscript.

I think that Figure 1 could be improved! Besides, in Table 1 the number of pure pixels used for calibration is much less than the number of pixels used for validation! Usually, it is the opposite when working with ML methods! Can you please elaborate on this subject?

Furthermore, the values in Table 1 are not in agreement with Figure 1, where the dominant colour is yellow (non-rice cal). In this table, only 4800 pure pixels were considered for calibration in both areas.

Author Response

Response to Reviewer Comments

The topic of the manuscript is interesting, and it is well structured. Models are well explained, and results are consistent with results already obtained by other authors. I suggest an English editing by a native speaker in order to improve the manuscript.

Response: Thank you very much for your great work. According to the comments, the authors amended the relevant part of the manuscript. The responses to the comments were given below.

Point 1: I think that Figure 1 could be improved! Besides, in Table 1 the number of pure pixels used for calibration is much less than the number of pixels used for validation! Usually, it is the opposite when working with ML methods! Can you please elaborate on this subject?

Response: Figure 1 has been re-done. Indeed, in the application of machine learning algorithms, if the sample and test samples are obtained in the same way, this rule (pixels used for calibration is much more than the number of pixels used for validation) will generally be followed. However, in the last review, a reviewer suggested “the itself model is applied to all pixels over the landscape and hence the verification dataset should be sampled from all pixels.” So we re-selected the validation samples according to the provided method. First, 800 pure pixels, including 200 rice pixels and 600 non-rice pixels, were identified by pure pixel extraction algorithm in both study areas. Then, all pixels in 7×7 square grid window centered on these pure pixels were used to evaluate the model performance. Therefore, the training samples and validation samples are selected in different ways and the number of validation samples is much larger than training samples.

Now, combining the suggestions of the two reviewers, we have re-extracted the data samples for accuracy assessment. We have clearly clarified this in the revision as below:

“Classification model based on training samples is applied to all pixels over the landscape and hence the verification dataset should be sampled from all pixels. Meanwhile, in order to avoid the spatial autocorrelation of validation samples, we randomly selected samples in the study area by setting a minimum distance (490 m in this article) between these sample points. This was done by the “Fishnet” function of ArcGIS software v10.3. After manual correction based on Google Earth Online image, 1575 pixels in Banan District and 1566 pixels in Zhongxian County were generated for the validation of resultant paddy rice map. Table 1 shows the sample division of the two study areas.”

Point 2: Furthermore, the values in Table 1 are not in agreement with Figure 1, where the dominant colour is yellow (non-rice cal). In this table, only 4800 pure pixels were considered for calibration in both areas.

Response 3: First, 800 pure pixels, including 200 rice and 600 non-rice grids, were identified in both Banan District and Zhongxian County (see “Non-rice val” and “Rice val” sites in Fig.1). Then, all pixels in 7×7 square grid window centered on these pure pixels were used to evaluate the model performance. Because some of the selected verification samples are clustered together and cannot be displayed clearly, we chose to display only 800 pure pixels (including 200 rice and 600 non-rice grids) in Figure 1.

Reviewer 2 Report

First of all, I appreciate the efforts that the authors have put in to change the methodology. Good job! Now, I see another issue in the methods, and hence I suggest a major revision:

The basic problem now is with this new scheme of selecting all 49 pixels ('Then, all pixels in 7×7 square grid window centered on these pure pixels were used to evaluate the model performance.'). That is, you are selecting a cluster of pixels, all of them next to each other. This violates the 'independence' assumption for validation samples. Pls refer to this seminal paper: Congalton, Russell G. "A review of assessing the accuracy of classifications of remotely sensed data." Remote sensing of environment 37.1 (1991): 35-46. Also, in the textbook: Campbell, James B., and Randolph H. Wynne. Introduction to remote sensing. Guilford Press, 2011 (5th edition), pg 402, there is a whole section ‘spacing of observations’. 

The best way, of course, is to select just one pixel. And make sure that the selected pixels are at a minimum distance from each other, broadly keeping in mind that we do not want validation samples to be spatially autocorrelated. In general, the validation proceduce has to be very well thought-out and designed.

Secondly, it is generally a bad idea to use another satellite-derived map as the reference data, instead of actual ground data. Due to the fact that both are drawn from the same data source (satellite spectral data), there is a high chance that both may have errors in the same direction (eg, light green forest vegetation being classified as paddy in both). So, you tend to get (wrong) high-accuracy figures. This should be discussed at length in the discussions; should form at least one big discussion para. You should explain in detail what the class-level accuracy of the reference map was (you just say that the overall accuracy was over 90%). What was the accuracy for paddy crop in the reference map? Also, some more details about the methodology used to generate this map.

I think in Dong et al 2016 (the paper I suggested last time), the entire AOI was treated as one unit of validation, so ok...

I can comment on the results and the discussions after the authors have given me a reply on the above comments.

Also some minor comments in remotesensing-784059-peer-review-v1_comments.pdf (attached).

Author Response

Response to Reviewer Comments

First of all, I appreciate the efforts that the authors have put in to change the methodology. Good job! Now, I see another issue in the methods, and hence I suggest a major revision:

Response: Thank you very much for your great work. According to the comments, the authors amended the relevant part of the manuscript. The responses to the comments were given below.

Point 1: The basic problem now is with this new scheme of selecting all 49 pixels ('Then, all pixels in 7×7 square grid window centered on these pure pixels were used to evaluate the model performance.'). That is, you are selecting a cluster of pixels, all of them next to each other. This violates the 'independence' assumption for validation samples. Pls refer to this seminal paper: Congalton, Russell G. "A review of assessing the accuracy of classifications of remotely sensed data." Remote sensing of environment 37.1 (1991): 35-46. Also, in the textbook: Campbell, James B., and Randolph H. Wynne. Introduction to remote sensing. Guilford Press, 2011 (5th edition), pg 402, there is a whole section ‘spacing of observations’. The best way, of course, is to select just one pixel. And make sure that the selected pixels are at a minimum distance from each other, broadly keeping in mind that we do not want validation samples to be spatially autocorrelated. In general, the validation procedure has to be very well thought-out and designed.

I think in Dong et al 2016 (the paper I suggested last time), the entire AOI was treated as one unit of validation, so ok...

Response 1: Thanks for your suggestion. In order to “make sure that the selected pixels are at a minimum distance from each other”, we have re-extracted the data samples for accuracy assessment. We have clearly clarified this in the revision as below:

“Classification model based on training samples is applied to all pixels over the landscape and hence the verification dataset should be sampled from all pixels. Meanwhile, in order to avoid the spatial autocorrelation of validation samples, we randomly selected samples in the study area by setting a minimum distance (490 m in this article) between these points. This was done by the “Fishnet” function of ArcGIS software v10.3. After manual correction based on Google Earth Online image, 1575 pixels in Banan District and 1566 pixels in Zhongxian County were generated for the validation of resultant paddy rice map. Table 1 shows the sample division of the two study areas.”

In fact, Dong et al. 2016 did not treat the entire AOI as one unit of validation, but generated all pixels under the AOI for model validation. Refer to line 16 of the third paragraph of Chapter 2.5 or Table 1 as below:

In our previous manuscript, the sentence of verification samples selection "First, 800 pure pixels, including 200 rice and 600 non-rice grids, were identified in both Banan District and Zhongxian County (see" Non-rice val "and" Rice val "sites in Fig. 1)” may cause some misunderstandings "These were identified by the stratified random sampling technique, right? ”. In fact, 800 pure pixels were identified by pure pixel sample selection algorithm in both Banan District and Zhongxian County. Then, all pixels in 7×7 square grid window centered on these pure pixels were used to evaluate the model performance. The land cover area under the 7×7 square grid window centered on pure pixels is equivalent to the “pure land cover AOIs” proposed by Dong.

Point 1: Secondly, it is generally a bad idea to use another satellite-derived map as the reference data, instead of actual ground data. Due to the fact that both are drawn from the same data source (satellite spectral data), there is a high chance that both may have errors in the same direction (eg, light green forest vegetation being classified as paddy in both). So, you tend to get (wrong) high-accuracy figures. This should be discussed at length in the discussions; should form at least one big discussion para. You should explain in detail what the class-level accuracy of the reference map was (you just say that the overall accuracy was over 90%). What was the accuracy for paddy crop in the reference map? Also, some more details about the methodology used to generate this map.

Response 2: Thanks for your suggestion.

  1. Generally, this land-use map is generated using the visual interpretation approach and ground survey based on high-resolution images (0.2 m-2.5m). The Chinese government launched the second national land survey from 2007 to 2009. After 2010, a nationwide land change survey was conducted annually. There is no detailed introduction in English version, the Chinese version can refer at “http://www.gov.cn/zhengce/content/2008-03/28/content_2417.htm” “http://www.doc88.com/p-983614011427.html”(Technical Regulations for the Second National Land Survey:TD/T 1014-2007).In fact, the accuracy of over 90% is a conservative estimate.

  1. This type of data has been used as reference data in some studies [1-3]. For example, in “Zhang, M.; Lin, H.; Wang, G.X.; Sun, H.; Fu, J. Mapping Paddy Rice Using a Convolutional Neural Network (CNN) with Landsat 8 Datasets in the Dongting Lake Area, China. Remote Sens-Basel 2018, 10.”. “ training samples for each land-cover type based on the LULC map of Hunan Province (2016) and Google Earth images (2016) were selected randomly.”

[1] Liu, W.P.; Henneberry, S.R.; Ni, J.P.; Radmehr, R.; Wei, C.F. Socio-cultural roots of rural settlement dispersion in Sichuan Basin: The perspective of Chinese lineage. Land Use Policy 2019, 88, doi:10.1016/j.landusepol.2019.104162.

[2]Yu, G.M.; Yu, Q.W.; Hu, L.M.; Zhang, S.; Fu, T.T.; Zhou, X.; He, X.L.; Liu, Y.A.; Wang, S.; Jia, H.H. Ecosystem health assessment based on analysis of a land use database. Appl Geogr 2013, 44, 154-164, doi:10.1016/j.apgeog.2013.07.010.

[3]Zhang, M.; Lin, H.; Wang, G.X.; Sun, H.; Fu, J. Mapping Paddy Rice Using a Convolutional Neural Network (CNN) with Landsat 8 Datasets in the Dongting Lake Area, China. Remote Sens-Basel 2018, 10, doi:10.3390/rs10111840.

  1. The land use map is only used for visual comparison (i.e., qualitative evaluation) in our study. Similarly, many scholars compare the remote sensing classification products generated in their articles with other published products. For example, Dong et al. (2016) also compared the resultant paddy rice maps with MODIS and Landsat 7 based paddy rice maps in a few case regions [4] (See Figure 8 in [4] as below). Also, Figure 7 in “Zhang T, Tang H. A comprehensive evaluation of approaches for built-up area extraction from landsat oli images using massive samples[J]. Remote Sensing, 2019, 11(1): 2.”; Figure 14 in “Zhang, X.; Wu, B.; Ponce-Campos, G.E.; Zhang, M.; Chang, S.; Tian, F. Mapping up-to-Date Paddy Rice Extent at 10 M Resolution in China through the Integration of Optical and Synthetic Aperture Radar Images. Remote Sens.2018, 10, 1200.”. As commented in the manuscript “remotesensing-675608”, “It would be better to show a difference map between the classified and the reference map to give a better visual quality assessment of the classifiers”. So, in our article, “in order to give a better visual comparison, intersect and erase calculations were performed using ArcGIS on the reference land use map and our classification result.”

[4]Dong, J.; Xiao, X.; Menarguez, M.A.; Zhang, G.; Qin, Y.; Thau, D.; Biradar, C.; Moore III, B. Mapping paddy rice planting area in northeastern Asia with Landsat 8 images, phenology-based algorithm and Google Earth Engine. Remote Sens. Environ. 2016, 185, 142-154

Point 3: I can comment on the results and the discussions after the authors have given me a reply on the above comments.

Response 3: Thank you very much. We are looking forward to your reply.

Point 4: Also some minor comments in remotesensing-784059-peer-review-v1_comments.pdf (attached).

Response 4: Thanks for your advice, the authors have revised accordingly.

Round 2

Reviewer 2 Report

I trust that the authors have gone through all my previous recommendations, thought about all of them, and acted on them.

So, as I understand now, the authors have 1575 pixels in Banan District and 1566 pixels in Zhongxian County for validation. These were manually classified by looking at Google Earth images. If this is so, a separate section 2.2.3 should read:

2.2.3: Validation data: Validation data for this study was assembled by manual inspection of Google Earth satellite images. First, we randomly distributed a set of points (1575 pts in Banan, 1566 in Zhongxian), with a minumum spacing of 490m between these points. Then, these points were classified as Rice / non-Rice by manual inspection via the Google earth interface (imagery provided by TerraMetrics, CNES,..). Points that had very ambiguous land covers were omitted.

And of course, section 2.2.2 should be 'Training samples'.

Thus, you make it 100% clear to the reader what your validation data was. As it is a very important part of a remote sensing work, a well-written validation section is very important!

Author Response

Response to Reviewer Comments

I trust that the authors have gone through all my previous recommendations, thought about all of them, and acted on them.

Response: Thank you very much for your great work. According to the comments, the authors amended the relevant part of the manuscript. The responses to the comments were given below.

Point 1: So, as I understand now, the authors have 1575 pixels in Banan District and 1566 pixels in Zhongxian County for validation. These were manually classified by looking at Google Earth images. If this is so, a separate section 2.2.3 should read:

2.2.3: Validation data: Validation data for this study was assembled by manual inspection of Google Earth satellite images. First, we randomly distributed a set of points (1575 pts in Banan, 1566 in Zhongxian), with a minumum spacing of 490m between these points. Then, these points were classified as Rice / non-Rice by manual inspection via the Google earth interface (imagery provided by TerraMetrics, CNES,..). Points that had very ambiguous land covers were omitted.

And of course, section 2.2.2 should be 'Training samples'.

Thus, you make it 100% clear to the reader what your validation data was. As it is a very important part of a remote sensing work, a well-written validation section is very important!

Response 1: Thank you very much for your great work, the authors have revised accordingly.

Point 2: Minor points in attached remotesensing-784059-peer-review-v2_comments.pdf

Response 2: Thanks for your advice, the authors have revised accordingly.

This manuscript is a resubmission of an earlier submission. The following is a list of the peer review reports and author responses from that submission.

Round 1

Reviewer 1 Report

The topic of the manuscript is quite interesting. Also, the manuscript is well structured, and the language is clear. Models are well explained, and results are consistent with results already obtained by other authors. However, some minor English editing is required (some lines where problems were detected are 12, 67, 90, 114, 143, 189, 482 and 490).

Line 12 – has not been widely

Line 67 – have been usually used

Line 90 – applied by the RS community

Line 114 – study areas the Banan District and the Zhongxian County

Line 143 – Sentinel-2 bands

Line 189 – cells in black

Line 482 – could explain why

Line 490 – might be a result of datasets

Besides, can you please explain the red numbers in Figure 4? In this figure’s legend shouldn’t w be 5 instead of 7?

After, these minor corrections I believe that the manuscript will be ready for publishing.

Reviewer 2 Report

The relevance of the research is not in doubt. The introduction, methods and results are described in a fairly complete and clear manner.

It is original to analyze the possibility of transferring the developed approaches to another region. Although the results obtained are generally predictable.

There are no big comments to the work. Among minor remarks, we will point out the following:

In figure 1 satellite images can be removed as they do not carry any special load. Fig. 2 is better to remove from the manuscript. The average value of the class "non-rice" does not make much sense. 457 line of manuscript. The authors' desire for "more homogeneous and accurate images" is controversial. After all, the presence of separate pixels of another class is not necessarily a classification error. It would be good to mention it also in the manuscript.

Reviewer 3 Report

This paper compares several machine learning based techniques for the classification of rice paddy in two regions of China. The writing is good and easy to follow. A good amount of solid work has been done to compare several models, though it could be presented better. I recommend accepting the paper, but with major revisions.

The sentence construction and usage of english through this paper has to be refined quite a bit before its publication. Many sentences need polishing, even though their meaning is somewhat clear. The same goes for some subsection headings. Please consider revising the paper thoroughly, preferably taking help from an english native speaker. I give some examples below to illustrate my point, but this set is just illustrative (ie, not a complete listing of the issues):

  • “As one of popular deep learning algorithms, Convolutional Neural Networks (CNNs) have recently carried out the excellent performance in computer vision such as ImageNet contest [45]” should be “One of the popular deep learning algorithms, Convolutional Neural Networks (CNNs), had exhibited excellent performance levels in computer vision related contests such as ImageNet [45]”
  • “Other disadvantages like floods and droughts, soil degradation, shifting and economic restructuring also should not be overlooked in rice production” should be “Other adverse factors such as droughts, soil degradation and economic restructuring should not be overlooked when studying rice production patterns.”
  • L479: “However, the data used in the article have lower spectral resolution, lower spectral continuity, and the inputs have leaping fluctuations in the transition between the spectral index and the spectral band.” Not sure what “leaping fluctuations” mean. Maybe rewrite as “However, the data used in that article have lower spectral resolution, lower spectral continuity, and the inputs have leaping fluctuations in the transition between the spectral index and the spectral band
  • L207: “CNNs have recently become a popular Deep Neural Networks and carried out great success in computer vision such as ImageNet.” > CNNs have recently become a popular version of Deep Neural Networks, and have achieved good success in computer vision algorithm intercomparison platforms such as ImageNet”

General/introduction: There have been quite some previous efforts that have attempted to classify ???. There should be a paragraph that explicitly addresses this; starts with the sentence: “There have been quite a few (or many) previous point cloud classification efforts where neighbourhood information was leveraged…’. The para should conclude with a sentence like: “The current paper is different from the above set of studies in the sense that…” (or something similar).

When presenting results of OA and F1 score, it is enough to state numbers as two decimal places. For example, in the abstract, ‘result with OA of 0.9269…’  becomes ‘result with OA of 0.93…’. This is much more readable, and does not affect the conclusions of the paper. Pls correct in all instances.

Overall, the methods section of this paper is not done well and is very difficult to follow. A careful re-writing is needed, so that 1) Unnecessary info is omitted, 2) References are added to std. textbooks, papers.

Some specific comments:

L168: This is not the proper way to introduce the land cover database. The sentence should be something like this: ‘For estimates of land cover, we used the Chongqing Land Change Investigation Database (1-2 references). This database was constructed using… The reported accuracy was ???%. 

L168-202: This whole section is quite difficult to follow, please improve it. I guess you are trying to explain the patch concept used in the context of CNNs, you have to do a better job of it.

Also is section 2.2.2 (L167) about the ‘reference data’ (ground truth)? Then, you have to explicitly state it. If so, rename section as ‘reference data collection’ or something. Also says 1-2 sentences in the section, that using high-res imagery & the LC map was the best option available. Also give refs to some papers that have used similar... 

L205-257: Do we need almost 1.5 pages explaining CNNs? As I understand, you have reproduced pretty-much standard theory of CNNs. If so, please condense to a short paragraph, with 1-2 references to some standard textbooks.

For example, the steps could be made bulleted points. Another question is whether this whole procedure can be explained in much simpler terms. 

L 258, section 3.2 (CNN classifiers). Many details in this section are unclear (including fig 5). For example, how 2D pixel data is transformed to get 1D input data for Conv-1D classifier is unclear. In figure 5 too, many terms such as pool_length and activation are unclear. Pls revise this entire section, omit unnecessary details (and add in refs to standard textbooks).

Also pls see attachment: rs-713889-o-notrack_reviewComments.pdf for more.

Round 2

Reviewer 3 Report

Now, the paper is clearer after much modifications. As I see it, there are quite some flaws in the methods (see below). Hence, I suggest that the paper is rejected, so that the authors get substantial time to work on it.

Now that I understand what has been done better, I think that the whole section on the selection of pure pixels is not needed, and deviates from the main content of the paper. In fact, the whole section can be summed up by the sentences: “Consider a wxw square grid window such that the current pixel in consideration is at the center of it. The current pixel is considered 'pure' is all wxw pixel in the grid have the same land cover category as the current pixel (at the center).” I think that too much space and effort has gone into details of how this was implemented, which is unnecessary for a paper in a scientific paper (ie, we just need well-defined steps). I also suspect that there are better and ready-made ways to implement it (for example, Arcmap has Focal Statistics, see https://desktop.arcgis.com/en/arcmap/10.3/tools/spatial-analyst-toolbox/focal-statistics.htm. This can be easily used to find your ‘pure pixels’ and is highly optimized. I am pretty sure there are similar methods in GDAL/QGIS). Yes, you do have a rather clever way of identifying the pure pixels, but the description of this algorithm is best fitted for the appendix, or maybe another avenue.

It is a very big problem that no references to land cover are available. Is this product published by one of the national agencies of China? If so, there should be a website for it (even though it is in chinese). Putting in such a dataset makes it almost impossible for someone later to replicate your study, which is quite bad (for a scientific study). Please find and put in a reference.

Another major issue I see with this paper is that you have selected only a subset of pixels from the whole area (ie, pixels that are at least 30m or so away from the paddy field edge) and these were used for verification/validation of the model too. This is wrong: the itself model is applied to all pixels over the landscape and hence the verification dataset should be sampled from all pixels. For example, consider a pixel which was at the edge of a paddy field, which was 100% paddy. It should be classified as paddy...but is it? We never know. Also, there could be vegetation at the edge of paddy fields (that you exclude from your verification data) that could have been classified as paddy. Another consequence of this is that we cannot say that the accuracy of maps in Fig 6 is not 95% or whatever you claim in your table. It has unknown accuracy (because the model was used for pixels that it was not trained for, or validated for). I hope you understand the issue. See this paper for a similar work where they used stratified random sampling to get the validation data: Dong, Jinwei, et al. "Mapping paddy rice planting area in northeastern Asia with Landsat 8 images, phenology-based algorithm and Google Earth Engine." Remote sensing of environment 185 (2016): 142-154.

Also, validation by comparing to a land use map is not a good idea (even if the map is 90% accurate). There is high probability that they make the same mistakes at the same place (hence your accuracy will be 100% for those cases).

The quality of the figures are still quite low. I cannot see any red or yellow in figure 5. The colors should be always mentioned and labelled in any figure. Also pls split fig6 into two (6 & 7).